# Investigation of Selected Prevalence Factors Associated with EHV-2 and/or EHV-5 Infection in Horses with Acute Onset of Fever and Respiratory Signs

**DOI:** 10.3390/v17050612

**Published:** 2025-04-25

**Authors:** Kaitlyn James, Duane E. Chappell, Bryant Craig, Chrissie Pariseau, Cara Wright, Philip van Harreveld, Samantha Barnum, Nicola Pusterla

**Affiliations:** 1Department of Obstetrics, Gynecology and Reproduction Biology, Massachusetts General Hospital, Boston, MA 02114, USA; kaitlynej@gmail.com; 2Merck Animal Health, Rahway, NJ 07065, USA; duane.chappell@merck.com (D.E.C.); chrissie.pariseau@merck.com (C.P.); cara.wright@merck.com (C.W.); philip.van.harreveld@merck.com (P.v.H.); 3Department of Medicine and Epidemiology, School of Veterinary Medicine, University of California, Davis, CA 95616, USA; smmapes@ucdavis.edu

**Keywords:** EHV-2, EHV-5, respiratory infection, nasal secretions, qPCR, prevalence factors

## Abstract

The purpose of this study was to determine any associations of EHV-2, EHV-5, and dual infection with EHV-2/-5 with demographic parameters, clinical signs, and coinfection with other common respiratory pathogens. Nasal swabs collected from 9737 horses were tested for EHV-2 and EHV-5, as well as EHV-1, EHV-4, EIV, *S equi*, ERAV, and ERBV, by qPCR. Clinical signs and demographic parameters were recorded, and prevalence factors were evaluated for significance regarding EHV-2 and/or EHV-5 infection. Out of the 9737 horses in this study, 17.8% tested EHV-2-positive (*n* = 1731), 15.8% tested EHV-5-positive (*n* = 1536), 33.4% tested positive for both viruses EHV-2/-5 (*n* = 3247), and 33.1% tested negative for both viruses (*n* = 3223). When comparing EHV-2 and/or EHV-5 infected horses to horses testing qPCR-negative for both viruses, horses infected with EHV-2 alone were more likely to be younger Thoroughbreds with a history of recent transportation, presenting with fever, and having a higher rate of coinfections with EHV-4 and *S. equi*. Horses infected with EHV-5 alone were less likely to be used for pleasure purposes, had fewer clinical signs, and were more likely to be coinfected with EHV-4. Horses dually infected with EHV-2 and EHV-5 were much more likely to be younger, used for competition, presenting with a fever, and coinfected with additional respiratory pathogens. It is apparent from the study population that horses infected with EHV-2 alone or in combination with EHV-5 had breed predilections, greater frequency of clinical signs, and a higher rate of coinfections with EHV-4, ERBV, and *S. equi*.

## 1. Introduction

Equine herpesvirus 2 (EHV-2) and equine herpesvirus 5 (EHV-5), collectively making up the equine gammaherpesvirus sub-family, have been associated with respiratory disease [1,2]. However, both viruses can also be found in clinically normal horses, confounding potential links to respiratory disease. Previous studies have shown that EHV-2 and EHV-5 infect horses within the first few months of life, and that life-long latency, a defining factor of many herpesviruses, is established. The detection of EHV-2 and EHV-5 can occur sporadically in respiratory and conjunctival swabs, as well as blood samples, but it is currently unknown as to whether this is due to the reactivation of latent infection or reinfection with different strains [1,3,4].

While no clinical diseases have been truly attributed to EHV-2 and -5 infection due to their presence in immunocompetent horses with no clinical signs, defining the clinical syndromes and potential for coinfection with these two viruses has been difficult. EHV-2 also demonstrates potential as a co-factor in bacterial respiratory infections due to *Rhodococcus equi* and has been implicated in the transactivation of EHV-1 and EHV-4 due to its ability to modulate host immune system responses and activate early genes [5,6,7,8]. A recent study suggested that EHV-5 may be linked to equine multinodular pulmonary fibrosis (EMPF), as EHV-5 was found in affected lung tissues when compared to unaffected lung tissues [9]. The aim of this retrospective study was to determine any associations of EHV-2, EHV-5, and dual infection with EHV-2/-5 with demographic parameters, clinical signs, and coinfection with other common respiratory pathogens.

## 2. Materials and Methods

Nasal secretions were collected from 9737 horses with clinical signs of respiratory disease between September 2012 and December 2024. The samples were submitted by veterinarians enrolled in a voluntary equine respiratory biosurveillance program. During this time frame, there were as many as 324 clinics enrolled in the program across 45 states. Nasal secretions were tested for EHV-2 and EHV-5 and other respiratory pathogens (EHV-1, EHV-4, equine influenza virus (EIV), *S. equi* subspecies *equi* (*S. equi*), and equine rhinitis A and B viruses (ERAV, ERBV)) using qPCR, as previously reported [10,11]. Horses were classified as having EHV-2 infection alone, EHV-5 infection alone, dual EHV-2/-5 infection, or no EHV-2/-5 infection.

Demographic frequency tables for the clinical outcomes were initially created to determine the similarities or differences in the populations. These demographic factors included breed, sex, age, and use. ‘Breed’ was divided into Quarter horse, Warmblood, Thoroughbred, Paint horse, Arabian, Draft horse, Pony/miniature, and other; ‘age’ was analyzed as a five-year increment categorical variable, with horses less than 1 year of age considered as the reference category; ‘sex’ was categorized into male (gelding and stallion) and female; and ‘use’ of animal was categorized into competition animals, pleasure animals, breeding animals, other or unknown uses. The reference for the ‘breed’ category was chosen based on greatest frequency of horses in the sample (in this case, Quarter horse), the reference for the ‘use’ category was arbitrarily chosen as competition, and the reference for the ‘sex’ variable was chosen arbitrarily as female.

Multivariable logistic regression models, with EHV-2/-5 negative horses serving as the reference for the other infection categories, were created to assess individual covariate associations with the three clinical outcomes compared to horses negative for both viruses. Risk factors that were considered significant in the univariate analysis (*p* < 0.10) were included in an adjusted multivariable logistic regression model. All statistical analyses and data management tasks were performed in Stata/SE 18.0 (StataCorp LLC, College Station, TX, USA).

## 3. Results

Data on breed, age, use, clinical signs, and possible coinfections were available for the majority of the 9737 horses (Table 1 and Table 2). The most prominent breed was Quarter horse (37%), the median age was 7 years (IQR 2–13 years), and horses were primarily used for competition or pleasure purposes (76.4%). The study population was 17.8% qPCR-positive for EHV-2 alone, 15.8% qPCR-positive for EHV-5 alone, and 33.4% qPCR-positive for dual infection with EHV-2/-5. About 10% of the horses tested qPCR-positive for EHV-4, EIV, or *S. equi*, while less than 6% of them tested qPCR-positive for ERBV or EHV-1.

The frequencies of demographic factors and outcome status are presented in Table 1. Draft horses had the lowest percentage of qPCR-positive horses in the EHV-2 alone group (56/1731; 3.2%) and in the EHV-2/-5 combined group (58/3247; 1.8%). For EHV-5 qPCR-positive results, the lowest frequency was observed in pony/miniature horses (48/1536; 3.1%). Almost half of the Thoroughbreds (693/1515; 46%) in the study sample were dually infected with EHV-2/-5. There were no sex differences in horses positive for either virus individually, while males were less likely to be dually infected. Younger horses (under 1 year of age) were less likely to be positive for EHV-5 (121/1566; 8%) and more likely to be positive for EHV-2 alone (298/1566; 19%) and dual infections with EHV-2/-5 (947/1566; 60%). There was no difference in recent history of transportation regarding EHV-2 or -5 infection.

Horses infected with EHV-2 alone or in combination with EHV-5 showed a higher rate of clinical signs such as fever, nasal discharge, ocular discharge, and coughing when compared to horses infected with EHV-5 alone and EHV-2/-5 qPCR-negative horses (Table 2). Amongst EHV-2- and/or EHV-5-infected horses, the rates of coinfections with EHV-1 (0.9–1.6%) and EIV (10.1–10.9%) were similar. For EHV-4 and ERBV, the highest frequency of detection was found in the EHV-2/-5 dually infected horses (486/3247, or 15.0%, for EHV-4 and 286/3247, or 8.8%, for ERBV). For horses testing qPCR-positive for *S. equi*, the highest frequency was found in the dual EHV-2/-5 infection group (435/3247; 13.4%).

The results of the unadjusted and adjusted multivariate models for EHV-2, EHV-5, and dual EHV-2/-5 infection are presented in Table 3, Table 4 and Table 5. For EHV-2 qPCR-positive compared to EHV-2/-5 negative horses, there were various demographic associations. Thoroughbred horses were more likely to be positive for EHV-2 alone compared to Quarter horses (aOR 1.60 (95% CI 1.24, 2.09); *p* < 0.001; Table 5), while horses used for breeding purposes were more likely to be positive than horses used for competition (aOR 1.62 (95% CI 1.05, 2.49); *p* = 0.03). Younger horses were more likely to be EHV-2-positive than EHV-2/-5-negative horses. EHV-2-positive horses were more likely to have a recent history of transportation (aOR 1.34 (1.11, 1.62); *p* = 0.002) and to demonstrate a fever (aOR 1.76 (95% CI 1.34, 2.33); *p* < 0.001). They were also more likely to be coinfected with EHV-4 (aOR 1.36 (95% CI 1.03, 1.81); *p* = 0.03) and *S. equi* (aOR 1.81 (95% CI 1.34, 2.43); *p* < 0.001). For EHV-5 qPCR-positive horses, there were fewer associations with demographics and clinical signs compared to EHV-2-/-5 negative animals in the adjusted model. Arabian horses were less likely to be EHV-5-positive alone compared to Quarter horses (aOR 0.60 (95% CI 0.41, 0.90); *p* = 0.01), as were horses used for pleasure purposes versus competition (aOR 0.74 (95% CI 0.61, 0.91); *p* = 0.004). There was no strong association with age or clinical signs. Coinfection with EHV-4 was associated with EHV-5 alone versus horses negative for both EHV-2 and/or -5 (aOR 1.36 (95% CI 1.02, 1.81); *p* = 0.038). For horses with dual EHV-2/-5 infection and compared to EHV-2/-5 qPCR-negative animals, breed was a significant risk factor; Thoroughbreds were more likely to be positive compared to the reference Quarter horse breed (aOR = 1.45 (95% CI 1.15, 1.82); *p* < 0.001), while Warmbloods, Draft horses, Paint horses, and other breeds were less likely to be dually infected. Younger horses were more likely to be infected compared to older horses. Horses used for pleasure were less likely to be dually infected compared to competition horses (aOR = 0.75 (95% CI 0.63, 0.89); *p* = 0.001). Dually infected horses were more likely to display fever compared to EHV-2/-5 qPCR-negative horses (aOR 1.53 (95% CI 1.22, 1.92); *p* < 0.001). Horses qPCR-positive for EHV-4 (aOR 1.65 (95% CI 1.30, 2.10); *p* < 0.001), ERBV (aOR 2.04 (95% CI 1.38, 3.00); *p* < 0.001), and *S. equi* (aOR 1.90 (95% CI 1.46, 2.48); *p* < 0.001) were more likely to be dually infected with EHV-2/-5.

## 4. Discussion

The results of this study showed that EHV-5 infection alone was not associated with severe clinical disease. Infections with EHV-2 alone or in combination with EHV-5 were associated with greater frequency of nasal discharge, ocular discharge, cough, and fever. Further, EHV-2 alone or in combination with EHV-5 showed greater association with coinfections compared to horses infected with EHV-5 only. These observations indicate that EHV-2 may play a modulatory role in the complex pathogenicity of upper respiratory tract infections. Previous studies looking at the association of equine gammaherpesviruses with respiratory disease have shown inconsistent results with either a positive association or no association of EHV-2 and EHV-5 with respiratory infections [11,12,13,14,15,16,17]. Differences in populations and the number of study animals, as well as the pathogenicity of different strains of equine gammaherpesviruses, may relate to the observed differences, especially with regard to EHV-2 infection. The present study offers a unique insight into the possible role of equine gammaherpesviruses due to the high case load and the fact that it compares horses infected with EHV-2 and/or EHV-5 with uninfected horses that are diseased.

Various demographic differences were found among EHV-2 and/or EHV-5 qPCR-positive horses. These observations may relate to breed-, age- and use-specific differences in management and husbandry practices, the intensity of preventative protocols, and susceptibility levels. The present study showed that horses less than one year of age were more likely to test EHV-2 qPCR-positive alone or in combination with EHV-5. These results agree with previous studies showing a significant higher detection rate of EHV-2 in foals compared to adult horses and highlighting frequent transmissions from mares to foals by direct contact during the first months of life [3,13]. The age-related changes in EHV-2 and/or EHV-5 detection observed in the present study may relate to the reduced circulation of gammaherpesviruses in adult horses, protective immunity from previous infections, reduced reactivation, or cross-protection from other respiratory viruses. Work performed in mice has shown that previous viral infection with murine gammaherpesvirus can modulate the inflammatory responses and decrease susceptibility to a heterologous virus in a time-dependent manner [18].

From the unadjusted and adjusted multivariate models, the frequency of clinical signs, especially fever, ocular and nasal discharge and cough, was higher in horses infected with EHV-2 alone or in combination with EHV-5. This observation may relate to the age of the study animals, as younger animals were over-represented, an age group known to display increased susceptibility to respiratory pathogens [10]. While it is possible that EHV-2 alone or in combination with EHV-5 has a direct pathogenic impact on the respiratory apparatus, it is also possible that the increased clinical disease expression was caused by the increased frequency of coinfections with other respiratory pathogens. Little information is available regarding the effect of equine respiratory coinfections with regard to disease frequency and disease expression [19,20]. EHV-2 and, likely, EHV-5 have a number of host gene homologues that are expected to function in host modulation during infection [1]. It still remains to be proven if a cumulative immunomodulatory effect from dual EHV-2/EHV-5 infection can further compromise host immunity and increase susceptibility to other respiratory pathogens.

The importance of strain could not be assessed in the current study due to limitations in covariate collection. The importance of strain for EHV-2 has been well documented in certain horse populations, most notably in Iceland, with a particular variant causing induced syncytium formation in equine kidney cells [21]. While information on the genomic diversity of EHV-5 is lacking, work on human cytomegalovirus (human herpesvirus-5) and Epstein–Barr virus (human herpesvirus-4) suggests that genomic diversity may contribute to clinical manifestation [22,23]. The inability to stratify by strain type in the current study limits the discussion of whether strain, rather than singular infection status, is the driving factor of pathogenicity and the presence of clinical signs. In addition, the lack of a true negative control population limits the ability to compare horses with clinical signs to healthy horses. Future studies should focus on collecting samples from unaffected horses originating from the same farm as affected horses to control for any geographic, husbandry, or management practices that may be associated with infection.

## 5. Conclusions

In conclusion, the increased frequency of nasal discharge, ocular discharge, cough and fever associated with EHV-2 infection alone or in combination with EHV-5 could indicate a cumulative effect in the pathogenesis of these two equine gammaherpesviruses or an immune-modulatory effect increasing susceptibility to other pathogens. However, the lack of severe clinical disease associated with EHV-5 infection alone does not preclude the role of this virus in disease frequency and expression. While it is impossible to determine from this study sample the temporal infection status of these respiratory pathogens, future research should focus on the strain differences with singular infection of gammaherpesviruses and the potential for immunomodulatory effects of EHV-2 and EHV-5 infection in horses.

## Figures and Tables

**Table 1 viruses-17-00612-t001:** Demographic factors associated with EHV-2 and EHV-5 qPCR-positivity among 9737 horses from 2012–2024.

	Positive EHV-2 Only (*n* = 1731)	Positive EHV-5 Only (*n* = 1536)	Positive EHV-2 and EHV-5 (*n* = 3247)	Negative EHV-2 and EHV-5 (*n* = 3223)
**Breed**				
Quarter Horse (*n* = 3629)	653 (37.7%)	595 (38.7%)	1301 (40.1%)	1080 (33.5%)
Thoroughbred (*n* = 1515)	285 (16.5%)	217 (14.1%)	693 (21.3%)	320 (9.9%)
Warmblood (*n* = 987)	154 (8.9%)	189 (12.3%)	232 (7.1%)	412 (12.8%)
Paint (*n* = 388)	71 (4.1%)	64 (4.2%)	99 (3.0%)	154 (4.8%)
Arabian (*n* = 537)	100 (5.8%)	63 (4.1%)	175 (5.4%)	199 (6.2%)
Draft Horse (*n* = 293)	56 (3.2%)	50 (3.3%)	58 (1.8%)	129 (4.0%)
Pony/Miniature (*n* = 397)	81 (4.7%)	48 (3.1%)	120 (3.7%)	148 (4.6%)
Other Breed (*n* = 1991)	331 (19.1%)	310 (20.2%)	569 (17.5%)	781 (24.2%)
**Use**				
Competition (*n* = 3833)	642 (37.1%)	643 (41.9%)	1391 (42.8%)	1157 (35.9%)
Pleasure horse (*n* = 3603)	650 (37.6%)	554 (36.1%)	966 (29.8%)	1433 (44.5%)
Breeding (*n* = 477)	85 (4.9%)	73 (4.8%)	180 (5.5%)	139 (4.3%)
Other Use (*n* = 546)	97 (5.6%)	95 (6.2%)	220 (6.8%)	134 (4.2%)
Unknown (*n* = 1278)	257 (14.8%)	171 (11.1%)	490 (15.1%)	360 (11.2%)
**Sex**				
Female (*n* = 3509)	614 (35.5%)	533 (34.7%)	1276 (39.3%)	1086 (33.7%)
Male ^a^ (*n* = 5057)	929 (53.7%)	816 (53.1%)	1549 (47.7%)	1763 (54.7%)
Unknown (*n* = 1171)	188 (10.9%)	187 (12.2%)	422 (13.0%)	374 (11.6%)
**Age**				
Less than 1 (*n* = 1566)	298 (17.2%)	121 (7.9%)	947 (29.2%)	200 (6.2%)
1–5 (*n* = 2381)	406 (23.5%)	420 (27.3%)	872 (26.9%)	683 (21.2%)
6–10 (*n* = 2010)	360 (20.8%)	349 (22.7%)	455 (14.0%)	846 (26.2%)
11–15 (*n* = 1479)	257 (14.8%)	261 (17.0%)	334 (10.3%)	627 (19.5%)
16–20 (*n* = 936)	174 (10.1%)	159 (10.4%)	242 (7.5%)	361 (11.2%)
Over 20 (*n* = 683)	122 (7.0%)	111 (7.2%)	173 (5.3%)	277 (8.6%)
Unknown (*n* = 682)	114 (6.6%)	115 (7.5%)	224 (6.9%)	229 (7.1%)
**History of transport**				
Yes (*n* = 2673)	526 (30.4%)	430 (28.0%)	847 (26.1%)	870 (27.0%)
No (*n* = 6208)	1064 (61.5%)	982 (63.9%)	2088 (64.3%)	2074 (64.3%)
Unknown (*n* = 856)	141 (8.1%)	124 (8.1%)	312 (9.6%)	279 (8.7%)

^a^ Male includes geldings and stallions.

**Table 2 viruses-17-00612-t002:** Clinical signs and coinfection associated with EHV-2 and EHV-5 qPCR-positivity among 9737 horses from 2012–2024.

	Positive EHV-2 Only (*n* = 1731)	Positive EHV-5 Only (*n* = 1536)	Positive EHV-2 and EHV-5 (*n* = 3247)	Negative EHV-2 and EHV-5 (*n* = 3223)
**Clinical signs**				
Nasal discharge (*n* = 7140)	1316 (76.0%)	1064 (69.3%)	2519 (77.6%)	2241 (69.5%)
Ocular discharge (*n* = 2431)	458 (26.5%)	359 (23.4%)	918 (28.3%)	696 (21.6%)
Cough (*n* = 4252)	796 (46.0%)	666 (43.4%)	1499 (46.2%)	1291 (40.1%)
Fever, defined as ≥101.5 °F (*n* = 8339)	1546 (89.3%)	1272 (82.8%)	2842 (87.5%)	2679 (83.1%)
Limb edema (*n* = 987)	164 (9.5%)	158 (10.3%)	350 (10.8%)	315 (9.8%)
Anorexia (*n* = 5870)	1064 (61.5%)	944 (61.5%)	1882 (58.0%)	1980 (61.4%)
Lethargy (*n* = 7157)	1314 (75.9%)	1109 (72.2%)	2333 (71.9%)	2401 (74.5%)
**qPCR results of nasal secretions**				
Positive for EHV-1 (*n* = 124)	19 (1.1%)	14 (0.9%)	53 (1.6%)	38 (1.2%)
Positive for EHV-4 (*n* = 1012)	170 (9.8%)	135 (8.8%)	486 (15.0%)	221 (6.9%)
Positive for EIV (*n* = 1016)	175 (10.1%)	168 (10.9%)	327 (10.1%)	346 (10.7%)
Positive for *S. equi* (*n* = 1018)	224 (12.9%)	108 (7.0%)	435 (13.4%)	251 (7.8%)
Positive for ERBV (*n* = 519)	94 (5.4%)	54 (3.5%)	286 (8.8%)	85 (2.6%)

**Table 3 viruses-17-00612-t003:** Univariate analysis of demographic factors associated with EHV-2 and/or EHV-5 qPCR-positivity compared to horses negative for both viruses amongst 9737 horses from 2012–2024.

	Positive EHV-2 Only (RRR, 95%CI, *p*-Value)	Positive EHV-5 Only (RRR, 95%CI, *p*-Value)	Positive EHV-2 and EHV-5 (RRR, 95%CI, *p*-Value)
**Breed**			
Quarter Horse	Ref.	Ref.	Ref.
Thoroughbred	1.47 (1.22, 1.78); <0.001	1.23 (1.01, 1.50); 0.04	1.79 (1.54, 2.10); <0.001
Warmblood	0.62 (0.50, 0.76); <0.001	0.83 (0.68, 1.02); 0.07	0.47 (0.39, 0.56); <0.001
Paint	0.76 (0.57, 1.03); 0.07	0.75 (0.55, 1.03); 0.07	0.53 (0.41, 0.70); <0.001
Arabian	NS	0.57 (0.43, 0.78); <0.001	0.73 (0.59, 0.91); 0.005
Draft Horse	0.72 (0.52, 0.99); 0.048	0.70 (0.50, 0.99); 0.044	0.37 (0.27, 0.51); <0.001
Pony/Miniature	NS	0.59 (0.42, 0.83); 0.002	0.67 (0.52, 0.86); 0.002
Other Breed	0.70 (0.60, 0.82); <0.001	0.72 (0.61, 0.85); <0.001	0.60 (0.53, 0.69); <0.001
**Use**			
Competition	Ref.	Ref.	Ref.
Pleasure	0.82 (0.72, 0.93); 0.003	0.70 (0.61, 0.80); <0.001	0.56 (0.50, 0.63); <0.001
Breeding	NS	NS	NS
Other Use	1.30 (0.99, 1.73); 0.06	1.28 (0.96, 1.69); 0.088	1.37 (1.09, 1.72); 0.008
**Sex**			
Female	Ref.	Ref.	Ref.
Male ^a^	NS	NS	0.72 (0.63, 0.81); <0.001
**Age**			
Less than 1	Ref.	Ref.	Ref.
1–5	0.40 (0.32, 0.50); <0.001	NS	0.29 (0.22, 0.33); <0.001
6–10	0.29 (0.23, 0.35); <0.001	0.68 (0.53, 0.88); 0.004	0.11 (0.09, 0.14); <0.001
11–15	0.28 (0.22, 0.35); <0.001	0.69 (0.53, 0.90); 0.006	0.11 (0.09, 0.14); <0.001
16–20	0.32 (0.25, 0.42); <0.001	0.73 (0.54, 0.98); 0.03	0.14 (0.11, 0.18); <0.001
Over 20	0.30 (0.22, 0.39); <0.001	0.66 (0.48, 0.91); 0.01	0.13 (0.10, 0.91); 0.01
**History of transport**			
Yes	1.18 (1.03, 1.35); 0.01	NS	NS
No	Ref.	Ref.	Ref.

^a^ Male includes geldings and stallions. NS = not significant.

**Table 4 viruses-17-00612-t004:** Univariate analysis of clinical signs and coinfection associated with EHV-2 and EHV-5 qPCR-defined positivity compared to horses negative for both viruses amongst 9737 horses from 2012–2024.

	Positive EHV-2 Only (RRR, 95%CI, *p*-Value)	Positive EHV-5 Only (RRR, 95%CI, *p*-Value)	Positive EHV-2 and EHV-5 (RRR, 95%CI, *p*-Value)
**Clinical signs**			
Nasal discharge	1.44 (1.25, 1.65); <0.001	NS	1.58 (1.40, 1.77); <0.001
Ocular discharge	1.31 (1.14, 1.51); <0.001	NS	1.44 (1.29, 1.62); <0.001
Cough	1.34 (1.18, 1.52); <0.001	1.15 (1.01, 1.31); 0.03	1.37 (1.23, 1.52); <0.001
Fever, defined as ≥101.5 °F	1.46 (1.19, 1.78); <0.001	NS	1.38 (1.17, 1.62); <0.001
**qPCR results of nasal** ** secretions**			
Positive for EHV-1	NS	NS	NS
Positive for EHV-4	1.47 (1.19, 1.81); <0.001	1.30 (1.04, 1.63); 0.02	2.38 (2.01, 2.81); <0.001
Positive for EIV	NS	NS	NS
Positive for *S. equi*	1.75 (1.45, 2.12); <0.001	NS	1.82 (1.55, 2.15); <0.001
Positive for ERBV	2.11 (1.56, 2.85); <0.001	1.34 (0.95, 1.90); 0.09	3.55 (2.78, 4.55); <0.001

NS = not significant.

**Table 5 viruses-17-00612-t005:** Adjusted multivariate analysis of demographic and clinical factors associated with EHV-2- and/or -5-positive horses compared to horses negative for both viruses, as defined by qPCR. The data were adjusted for breed, use, age, history of transport, sex, clinical signs (nasal discharge, ocular discharge, cough, fever), and coinfections (EHV-4, *S. equi*, and ERBV).

	Positive EHV-2 Only (RRR, 95%CI, *p*-Value)	Positive EHV-5 Only (RRR, 95%CI, *p*-Value)	Positive EHV-2 and EHV-5 (RRR, 95%CI, *p*-Value)
**Breed**			
Quarter Horse	Ref	Ref	Ref
Thoroughbred	1.60 (1.24, 2.09); <0.001	NS	1.45 (1.15, 1.82); 0.001
Warmblood	NS	NS	0.66 (0.51, 0.86); 0.002
Paint	NS	NS	0.65 (0.45, 0.95); 0.03
Arabian	NS	0.60 (0.41, 0.90); 0.01	0.67 (0.50, 0.92); 0.01
Draft Horse	NS	NS	0.50 (0.31, 0.79); 0.003
Pony/Miniature	NS	NS	NS
Other Breed	0.75 (0.59, 0.97); 0.03	NS	0.67 (0.54, 0.83); <0.001
**Use**			
Competition	Ref	Ref	Ref
Pleasure	NS	0.74 (0.61, 0.91); 0.004	0.75 (0.63, 0.89); 0.001
Breeding	1.62 (1.05, 2.49); 0.03	NS	NS
Other Use	NS	NS	NS
**Age**			
Less than 1	Ref	Ref	Ref
1–5	0.49 (0.35, 0.69); <0.001	NS	0.29 (0.22, 0.39); <0.001
6–10	0.33 (0.24, 0.47); <0.001	0.68 (0.46, 0.99); 0.045	0.14 (0.10, 0.19); <0.001
11–15	0.37 (0.25, 0.53); <0.001	NS	0.14 (0.11, 0.19); <0.001
16–20	0.43 (0.29, 0.64); <0.001	NS	0.20 (0.14, 0.28); <0.001
Over 20	0.54 (0.35, 0.81); 0.003	0.61 (0.38, 0.99); 0.048	0.22 (0.15, 0.31); <0.001
**History of transport**			
Yes	1.34 (1.11, 1.62); 0.002	NS	NS
No	Ref	Ref	Ref
Unknown	NS	NS	NS
**Clinical signs**			
Fever, defined as ≥101.5 °F	1.76 (1.34, 2.33); <0.001	NS	1.53 (1.22, 1.92); <0.001
**qPCR results of nasal secretions**			
Positive for EHV-4	1.36 (1.03, 1.81); 0.03	1.36 (1.02, 1.81); 0.038	1.65 (1.30, 2.10); <0.001
Positive for *S. equi*	1.81 (1.34, 2.43); <0.001	NS	1.90 (1.46, 2.48); <0.001
Positive for ERBV	NS	NS	2.04 (1.38, 3.00); <0.001

## Data Availability

Data are available upon request due to privacy restrictions.

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
