# Peer review of "Investigation of Selected Prevalence Factors Associated with EHV-2 and/or EHV-5 Infection in Horses with Acute Onset of Fever and Respiratory Signs"

_viruses, 2025, doi:10.3390/v17050612_

Round 1
Reviewer 1 Report
Comments and Suggestions for Authors
EHV-2 and EHV-5 are a kind of ubiquitous viruses which have been isolated from horses with no clinical signs. Pathogenic significance of these viruses has been unclear. Recent studies suggested that these viruses might be involved in various diseases in horses. The present study is retrospective to determine any associations between EHV-2, EHV-5 and dual EHV-2/-5 infection with demographic factors, clinical factors, and coinfection with common respiratory pathogens. Experimental design is scientifically organized. The results of the present study will provide researchers with new insights. However, description of the results is complicated and difficult to understand. Specific comments are as follows.
L53-55: Considering movements of horses and the spread of infection, is it necessary to analyze geographical conditions? If geographical conditions are not needed to be analyzed, it is better to mention no relation between geography and EHV-2/-5 infection.
L57-58: Please write down which country and where you collected the samples.
L89-98: The authors described positive "Warmblood horses had the lowest percentage of positive results for EHV-2 (15.6%) while pony/miniature horses had the lowest percentage of positive results for EHV-5 (12.1%)". However, reading Table 1, the lowest percentage of positive results for EHV-2 was 5% (3.2% plus 1.8%) for Draft Horse. And, the lowest percentage of positive results for EHV-5 was 5.1% (3.3% + 1.8%) for Draft Horse. In this paragraph, percentages described in the text are different from those in Table1. The differences will confuse readers. Describe the results using numbers shown in Table 1. Otherwise, table should consist of numbers described in the text. One of the solutions is to add a fraction after the percentage. This will help readers to understand the meaning of percentage.
L99-107: The authors described that out of 1016 EIV qPCR-positive horses, 32% were dually infected with EHV-2/-5 (Table 2). Actually, the percentage shown in Table 2 is 10.1%. What is this discrepancy? Readers expect "32%" in Table 2. However, percentages in Table 2 are calculated using different methods. The same discrepancies are found in this paragraph. One of the solutions is that the authors add actual numbers such as 32% (327/1016), 55% (286/519) and so on.
L108-131: Descriptions in this paragraph are complicated because of discrepancies between the text and the tables. The authors wrote Thoroughbred horses were more likely to be positive for EHV-2 alone compared to Quarter horses (aOR 1.60 (95% CI 1.24, 2.09); p < 0.001), while horses used for breeding purposes were more likely to be positive than horses used for competition (p = 0.03). I cannot find these numbers "(aOR 1.60 (95% CI 1.24, 2.09); p < 0.001)" in Table 3. Where do these numbers come from? Why did not the authors use numbers shown in Tables 3 to 5 in this paragraph? The authors have to unify descriptions between the text and tables.
L121, 127: Although the authors wrote "There was no strong association with age or clinical signs" in L121, the authors also wrote "Younger horses were more likely to be infected compared to older horses" in L127. Which is right? The authors must add further explanation to these descriptions.
L146-149: The authors discussed geographic differences here. However, the authors did not show geographical information in the present manuscript. The authors have to show geographical information of the samples used in the present manuscript. And, specific place names should be described here.
Overall, the description of the results is insufficient, making it difficult to understand the discussion.
It is necessary to match the contents of the table showing the results with the text, and describe the results more carefully.
Author Response
EHV-2 and EHV-5 are a kind of ubiquitous viruses which have been isolated from horses with no clinical signs. Pathogenic significance of these viruses has been unclear. Recent studies suggested that these viruses might be involved in various diseases in horses. The present study is retrospective to determine any associations between EHV-2, EHV-5 and dual EHV-2/-5 infection with demographic factors, clinical factors, and coinfection with common respiratory pathogens. Experimental design is scientifically organized. The results of the present study will provide researchers with new insights. However, description of the results is complicated and difficult to understand. Specific comments are as follows.
L53-55: Considering movements of horses and the spread of infection, is it necessary to analyze geographical conditions? If geographical conditions are not needed to be analyzed, it is better to mention no relation between geography and EHV-2/-5 infection.
The goal of this study was not to determine geographic associations. Index cases all originated from the United States and were enrolled as part of a voluntary biosurveillance program. The program enrolled about more than 300 equine veterinary clinics. Further, as the reviewer mentioned, sport horses often travel long distances, further confounding the issue of geographic origin of the horse. Also, the number of index cases follows the frequency of horse distribution with greater submissions originating from states with a high number of horses.
L57-58: Please write down which country and where you collected the samples.
Additional information was added to explain the origin of the samples. Nasal swabs were collected by veterinarian enrolled in a voluntary biosurveillance program for respiratory pathogens and targeting horses with acute onset of fever and respiratory signs.
L89-98: The authors described positive "Warmblood horses had the lowest percentage of positive results for EHV-2 (15.6%) while pony/miniature horses had the lowest percentage of positive results for EHV-5 (12.1%)". However, reading Table 1, the lowest percentage of positive results for EHV-2 was 5% (3.2% plus 1.8%) for Draft Horse. And, the lowest percentage of positive results for EHV-5 was 5.1% (3.3% + 1.8%) for Draft Horse. In this paragraph, percentages described in the text are different from those in Table1. The differences will confuse readers. Describe the results using numbers shown in Table 1. Otherwise, table should consist of numbers described in the text. One of the solutions is to add a fraction after the percentage. This will help readers to understand the meaning of percentage.
The authors are thankful to the reviewer for pointing out the discrepant results between text and Table 1. The results were adjusted to reflect the numbers in the table. The fractions and percentages were added for each rubric.
L99-107: The authors described that out of 1016 EIV qPCR-positive horses, 32% were dually infected with EHV-2/-5 (Table 2). Actually, the percentage shown in Table 2 is 10.1%. What is this discrepancy? Readers expect "32%" in Table 2. However, percentages in Table 2 are calculated using different methods. The same discrepancies are found in this paragraph. One of the solutions is that the authors add actual numbers such as 32% (327/1016), 55% (286/519) and so on.
This is the same issue as above and has been addressed by reporting the fractions and percentages.
L108-131: Descriptions in this paragraph are complicated because of discrepancies between the text and the tables. The authors wrote Thoroughbred horses were more likely to be positive for EHV-2 alone compared to Quarter horses (aOR 1.60 (95% CI 1.24, 2.09); p < 0.001), while horses used for breeding purposes were more likely to be positive than horses used for competition (p = 0.03). I cannot find these numbers "(aOR 1.60 (95% CI 1.24, 2.09); p < 0.001)" in Table 3. Where do these numbers come from? Why did not the authors use numbers shown in Tables 3 to 5 in this paragraph? The authors have to unify descriptions between the text and tables.
The results presented in the text refer to Table 5 and only list the adjusted OR (aOR). The missing information has been added in the results to prevent any confusion.
L121, 127: Although the authors wrote "There was no strong association with age or clinical signs" in L121, the authors also wrote "Younger horses were more likely to be infected compared to older horses" in L127. Which is right? The authors must add further explanation to these descriptions.
Sentence in original L 121 (now L 136) refers to EHV-5 only qPCR-positive horses while sentence in original L 127 (now L143) refers to EHV-2/-5 dually infected qPCR-positive horses.
L146-149: The authors discussed geographic differences here. However, the authors did not show geographical information in the present manuscript. The authors have to show geographical information of the samples used in the present manuscript. And, specific place names should be described here.
Geographic differences refer to populations of horses from different continents. As the present study did not investigate location/state of the study horses, the term geographic was removed.
Overall, the description of the results is insufficient, making it difficult to understand the discussion.
It is necessary to match the contents of the table showing the results with the text, and describe the results more carefully.
The authors thank the reviewer for pointing to the difficulty to follow the results and the discrepancies with the tables. Since the groups were divided into EHV-2 only, EHV-5 only, EHV-2/-5 dual infection and no EHV-2/-5 infection, it was important to report the results in the same group order. The authors hope to have improved the clarity of the results by matching the results with the tables.
Reviewer 2 Report
Comments and Suggestions for Authors
The manuscript focused on characterizing the association of single-pathogen infections alone or dual infections caused by EHV-2 or EHV-5 in horses showing fever and/or respiratory symptoms with demographic factors (e.g., age, breed, gender, ……), infection caused by other common respiratory pathogens (e.g. EIV, ERAV, ERBV, ……). These authors used qPCR to detect EHV-2 and EHV-5 nasal swabs from 9737 horses to generate a series of Tables. The results showed that the increased frequency of nasal discharge, ocular discharge, cough, and fever associated with EHV-2 infection alone or in combination with EHV-5 may indicate a cumulative effect in the pathogenesis of these two equine gammaherpesviruses or an immune-regulating effect that increases susceptibility to other pathogens.
This research further confirms important pathogenesis of EHV-2 and EHV-5 in terms of clinical disease. They could not be neglected during diagnosing and therapying respiratory disease in horses. This manuscript inspired the future researches that should concentrate on immunoregulatory roles by EHV-2 and EHV-5. The comments are listed as below:
Line 15. “The purpose of this study was to characterize associations between demographic factors, clinical signs, and coinfections with EHV-2 and/or EHV-5 infection in horses with acute onset of fever and/or respiratory signs.” Could you clarify what you mean by that? May be revised furtherly? The authors should consider the sentence pattern.
Line 36. “Equine herpesvirus-2” should be corrected to “Equine herpesvirus 2”, and the same principle applies to the whole manuscript.
Line 38. “...... of clinical signs and disease outcomes”. The corresponding clinical signs and disease outcomes should be readily listed in this section.
Line 47-48. “Equine herpesvirus-2” should be corrected to “EHV-2”, because the abbreviated style has been generated at Line 36 when the EHV-2 appeared firstly.
Line 54-55. “……between EHV-2, EHV-5 and dual EHV-2/-5 infection with demographic factors, clinical factors, and coinfection with common respiratory pathogens.” The sentence is very difficult to understand.
This sentence may be revised to “……between EHV-2 infection as well as EHV-5 associated with demographic or clinical factors, and coinfection caused by common respiratory pathogens (e.g. EIV, EAV, …… and so on).
Line 57. “……9,737 equids……”. Are there some nasal swabs from donkeys and mules? If these samples only from horses, this expression should be revised.
Line 139. “……while dual infection with EHV-2/-5 and infection with EHV-2 alone
was associated……”. This sentence should be restructured to make it simple.
Author Response
The manuscript focused on characterizing the association of single-pathogen infections alone or dual infections caused by EHV-2 or EHV-5 in horses showing fever and/or respiratory symptoms with demographic factors (e.g., age, breed, gender, ……), infection caused by other common respiratory pathogens (e.g. EIV, ERAV, ERBV, ……). These authors used qPCR to detect EHV-2 and EHV-5 nasal swabs from 9737 horses to generate a series of Tables. The results showed that the increased frequency of nasal discharge, ocular discharge, cough, and fever associated with EHV-2 infection alone or in combination with EHV-5 may indicate a cumulative effect in the pathogenesis of these two equine gammaherpesviruses or an immune-regulating effect that increases susceptibility to other pathogens.
This research further confirms important pathogenesis of EHV-2 and EHV-5 in terms of clinical disease. They could not be neglected during diagnosing and therapying respiratory disease in horses. This manuscript inspired the future researches that should concentrate on immunoregulatory roles by EHV-2 and EHV-5. The comments are listed as below:
Line 15. “The purpose of this study was to characterize associations between demographic factors, clinical signs, and coinfections with EHV-2 and/or EHV-5 infection in horses with acute onset of fever and/or respiratory signs.” Could you clarify what you mean by that? May be revised furtherly? The authors should consider the sentence pattern.
In order to prevent any confusion, the sentence has been changed to “The purpose of this study was to determine any associations between EHV-2, EHV-5 and dual EHV-2/-5 infection with demographic factors, clinical factors, and coinfection with common respiratory pathogens.”.
Line 36. “Equine herpesvirus-2” should be corrected to “Equine herpesvirus 2”, and the same principle applies to the whole manuscript.
Many thanks for bringing up this issue, the authors have corrected the spelling of EHV-2 as suggested.
Line 38. “...... of clinical signs and disease outcomes”. The corresponding clinical signs and disease outcomes should be readily listed in this section.
Clinical signs relate to mild respiratory disease. The sentence has been modified.
Line 47-48. “Equine herpesvirus-2” should be corrected to “EHV-2”, because the abbreviated style has been generated at Line 36 when the EHV-2 appeared firstly.
The change has been made in the manuscript.
Line 54-55. “……between EHV-2, EHV-5 and dual EHV-2/-5 infection with demographic factors, clinical factors, and coinfection with common respiratory pathogens.” The sentence is very difficult to understand.
This sentence may be revised to “……between EHV-2 infection as well as EHV-5 associated with demographic or clinical factors, and coinfection caused by common respiratory pathogens (e.g. EIV, EAV, …… and so on).
The authors agree with the reviewer that the sentence is long and potentially hard to read. To add clarity to the aim of the study, the sentence has been changed to “The aim of this retrospective study was to determine any associations between EHV-2, EHV-5 and dual infection with EHV-2/-5 with demographic parameters, clinical signs, and coinfection with other common respiratory pathogens”.
Line 57. “……9,737 equids……”. Are there some nasal swabs from donkeys and mules? If these samples only from horses, this expression should be revised.
The term equids was switched to horses since no mules or donkeys were part of the study population.
Line 139. “……while dual infection with EHV-2/-5 and infection with EHV-2 alone
was associated……”. This sentence should be restructured to make it simple.
The sentence has been restructured to add clarity.
Round 2
Reviewer 1 Report
Comments and Suggestions for Authors
Answers to the comments are acceptable. The manuscript is revised adequately.